# SPECIALIZATION OF SUB-PATHS FOR ADAPTIVE DEPTH NETWORKS

## ABSTRACT

We present a novel approach to anytime networks that can control network depths instantly at runtime to provide various accuracy-efficiency trade-offs. While controlling the depth of a network is an effective way to obtain actual inference speed-up, previous adaptive depth networks require either additional intermediate classifiers or decision networks, that are challenging to train properly. Unlike previous approaches, our approach requires virtually no architectural changes from baseline networks. Instead, we propose a training method that enforces some sub-paths of the baseline networks to have a special property, with which the sub-paths do not change the level of input features, but only refine them to reduce prediction errors. Those specialized sub-paths can be skipped at test time, if needed, to save computation at marginal loss of prediction accuracy. We first formally present the rationale behind the sub-paths specialization, and based on that, we propose a simple and practical training method to specialize sub-paths for adaptive depth networks. Our approach is generally applicable to residual networks including both convolution networks and vision transformers. We demonstrate that our approach outperforms non-adaptive baseline residual networks in various tasks, including ImageNet classification, COCO object detection and instance segmentation.

## 1 INTRODUCTION

Modern deep neural networks provide state-of-the-art performance at high computational costs, and, hence, lots of efforts have been made to leverage those inference capabilities in resource-constrained systems, such as autonomous vehicles. Those efforts include compact architectures (Howard et al., 2017; Zhang et al., 2018; Han et al., 2020), network pruning (Han et al., 2016; Liu et al., 2019), weight/activation quantization (Jacob et al., 2018), knowledge distillation (Hinton et al., 2015), to name a few. However, those approaches provide static accuracy-efficiency trade-offs that are often tailored for worst-case scenarios, and, hence, the lost accuracy cannot be recovered even if more resources become available.

Adaptive networks such as anytime networks (Huang et al., 2018; Yu et al., 2018; Wan et al., 2020) attempt to provide runtime adaptability to deep neural networks by exploiting the redundancy in either depths or widths, as shown in Figure 1, or resolutions (Yang et al., 2020a). Dynamic networks (Wu et al., 2018; Li et al., 2021; 2020; Zhu et al., 2021) add additional control logic to the backbone network for input-dependent adaptation. However, these adaptive networks usually require auxiliary networks, such as intermediate classifiers or decision networks, which are challenging to train properly. Further, since adaptive networks have multiple sub-networks, embedded in a single neural network, training them incurs potentially conflicting training objectives for the sub-networks, resulting in worse performance than non-adaptive networks (Li et al., 2019).

In this work, we introduce a novel approach to anytime networks that is executable in multiple depths to provide instant runtime accuracy-efficiency trade-offs. Unlike previous adaptive depth networks, our approach does not require additional add-on networks or classifiers, and, hence, it can be applied to modern residual networks easily. While maintaining the structure of original networks, we train several sub-paths, or a sequence of residual blocks, of the network to have a special property, that preserves the level of input features, and only refines them to reduce prediction errors. At test time, these specialized sub-paths can be skipped, if needed, for efficiency at marginal loss of accuracy as shown in Figure-1 (right).

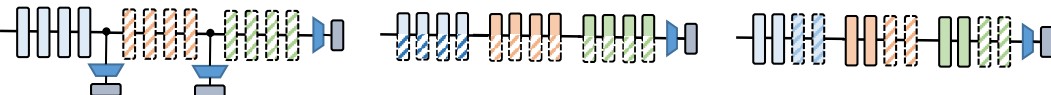

Figure 1: Anytime networks with (**left**) early-exit branches, (**middle**) adaptive widths, or channels, and (**right**) specialized sub-paths (ours). Dashed layers (or blocks) and channels can be skipped for instant accuracy-efficiency trade-offs at runtime.

The proposed sub-paths specialization is achieved by enforcing sub-networks with different depths to produce features with similar distributions for every spatial dimension. In Section 3, we formally discuss the rationale behind the sub-paths specialization and introduce a simple and practical training method for sub-paths specialization. In most previous adaptive networks, the total training time is linearly proportional to the number of supported sub-networks, and resolving potential conflicts between sub-networks is an important problem. In contrast, our approach does not try to resolve potential conflicts while jointly training many sub-networks. Instead, our training method exploits only two sub-networks for sub-paths specialization, and, at test time, those specialized sub-paths are exploited selectively to build many sub-networks of various depths. Therefore, the total training time is no greater than training two individual networks. Further, our approach with sub-paths specialization do not exploit specific properties of convolution neural networks (CNNs) or vision transformers, and, hence, is generally applicable to residual networks, including both CNNs and recent vision transformers.

In Section 4, we demonstrate that our adaptive depth networks with sub-paths specialization outperform counterpart individual networks, both in CNNs and vision transformers, and achieve actual inference acceleration in multiple tasks including ImageNet classification, COCO object detection and instance segmentation. To the best of authors' knowledge, this work is the first general approach to adaptive networks demonstrating consistent performance improvements for both CNNs and vision transformers.

## 2  RELATED WORK

**Adaptive Networks:** Anytime networks (Larsson et al., 2017; Huang et al., 2018; Hu et al., 2019; Wan et al., 2020) and dynamic networks (Wu et al., 2018; Li et al., 2021; Guo et al., 2019; Li et al., 2020; Yang et al., 2020a) have attracted lots of attention for their runtime adaptability. Most anytime networks have multiple classifiers that are connected to intermediate layers (Huang et al., 2018; Li et al., 2019; Fei et al., 2022). Training multiple classifiers is a challenging task and many anytime networks (Li et al., 2019; Zhang et al., 2019; Wan et al., 2020; Huang et al., 2018; Hu et al., 2019) exploit knowledge distillation to supervise intermediate classifiers using the last, or the best, classifier. Slimmable neural networks can adjust channel widths for adaptability and they exploit switchable batch normalization to handle multiple sub-networks with a single shared classifier (Yu et al., 2018; Yu & Huang, 2019b). While some dynamic networks (Li et al., 2021) extend anytime networks simply by adding input-conditioned decision gates at branching paths, a few dynamic networks (Wu et al., 2018; Veit & Belongie, 2018; Wang et al., 2018a) extend residual networks by applying block-level decision gates that determine if the block can be skipped. The latter approach is based on the thought that some blocks can be skipped on easy inputs. However, in these dynamic networks with adaptive depths, no formal explanation has been given why some blocks can be skipped for a given input. Therefore, users have no control over the depth of the sub-networks.

**Residual Blocks with Shortcuts:** Since the introduction of ResNets (He et al., 2016), residual blocks with shortcuts have received extensive attention because of their ability to train very deep networks, and have been chosen by many recent deep neural networks (Sandler et al., 2018; Tan & Le, 2019; Vaswani et al., 2017). Veit et al. (2016) argue that identity shortcuts make exponential paths and results in an ensemble of shallower networks. This thought is supported by the fact that removing individual residual blocks at test time does not significantly affect performance (Huang et al., 2016; Xie et al., 2020). Other works argue that identity shortcuts enable residual blocks to perform iterative feature refinement, where each block improves slightly but keeps the semantic of the representation of the previous layer (Greff et al., 2016; Jastrzebski et al., 2018). Our work build upon those views on residual blocks with shortcuts. We further extend them for adaptive depth networks by introducing a

novel training method that exploits the properties of residual blocks more explicitly for sub-paths specialization.

## 3   Sub-Paths Specialization for Adaptive Depth Networks

In this section, we first formally discuss the rationale behind the sub-paths specialization. Then, the details of training of sub-paths specialization is discussed.

### 3.1   Motivation and Overview

In typical residual networks such as ResNets, the $s$-th residual stage is consisted of $L$ identical residual blocks, which transform input features $\mathbf{h}^{s-1}$ additively and produce the output features $\mathbf{h}^s$, as shown in Equation 1. While a block with a residual function $F$ learns higher level features as traditional compositional networks (Simonyan & Zisserman, 2015), previous literature (Jastrzebski et al., 2018; Greff et al., 2016) demonstrates that a residual function also tend to learn a function that refines already learned features at the same feature level. The first and the second functions, respectively, are called *feature learning* and *feature refinement*. If a residual block mostly perform feature refinement while not changing the level of features, the performance of the residual network is not significantly affected by dropping the block at test time (Huang et al., 2016; Xie et al., 2020). However, in typical residual networks, most residual blocks tend to learn both functions, and, hence, random dropping of residual blocks at test time degrades the performance significantly. If some residual blocks can be trained to focus on one function while the remaining blocks are encouraged to focus on the other function, then we can safely skip blocks specialized to feature refinement to save computation at marginal loss of prediction accuracy, if needed for efficiency.

To this end, in our approach, residual functions of a residual stage are partitioned into two sub-paths, $\mathbf{F}_{base}$ and $\mathbf{F}_{skippable}$, as in Equation 1, and, during training, they are encouraged to focus more on respective functions, i.e., learning new level features and refining the learned features.

$$\mathbf{h}^{s-1} + \underbrace{\underbrace{F_1(\mathbf{h}^{s-1}) + F_2(\mathbf{h}_1^s) + ... + F_K(\mathbf{h}_{K-1}^s)}_{\mathbf{F}_{base}} + \underbrace{F_{K+1}(\mathbf{h}_K^s) + ... + F_L(\mathbf{h}_{L-1}^s)}_{\mathbf{F}_{skippable}}}_{=\mathbf{h}_{base}^s} = \mathbf{h}_{super}^s \quad (1)$$

Since the last half residual functions, or $\mathbf{F}_{skippable}$, is supposed to preserve the level of input features, $\mathbf{F}_{skippable}$ can be skipped for efficiency and the intermediate features, or $\mathbf{h}_{base}^s$, can be used as the input to the next stage. When the higher inference accuracy is required, $\mathbf{F}_{skippable}$ can be applied to $\mathbf{h}_{base}^s$ to produce more refined features $\mathbf{h}_{super}^s$. Since the feature representations $\mathbf{h}_{base}^s$ and $\mathbf{h}_{super}^s$ are at the same level, either $\mathbf{h}_{base}^s$ or $\mathbf{h}_{super}^s$ can be provided as an input $\mathbf{h}^s$ to the next $(s+1)$-th residual stage with little change in the feature distribution.

Figure 2 shows an residual stage, in which the last half blocks are specialized to preserve the level of input features. Since each such residual stage provides two alternative internal paths, if there are $N_s$ residual stages with specialized sub-paths, $2^{N_s}$ sub-networks with different accuracy-efficiency trade-offs become available at test time (Section 4.3).

### 3.2   Sub-Paths Specialization

Preserving the level of input features implies that two features representations $\mathbf{h}_{base}^s$ and $\mathbf{h}_{super}^s$ have similar distributions over input $\mathbf{X}$. We can enforce this during training by minimizing the Kullback-Leibler (KL) divergence between the two feature representations:

$$D_{KL}(\mathbf{h}_{super}^s || \mathbf{h}_{base}^s) \to 0 \quad (2)$$

It is noteworthy that enforcing Equation 2 also has the distillation effect of transferring the knowledge from $\mathbf{h}_{super}^s$ to $\mathbf{h}_{base}^s$. Therefore, at each residual stage, $\mathbf{h}_{base}^s$ is expected to learn more compact representation from $\mathbf{h}_{super}^s$. As a consequence of Equation 2, we can conjecture that residual functions in $\mathbf{F}_{skippable}$ is trained to have small magnitude:

$$\mathbb{E}_{\mathbf{x} \in \mathbf{X}}[F_{K+1}(\mathbf{h}_K^s) + ... + F_L(\mathbf{h}_{L-1}^s)] \to 0 \quad (3)$$

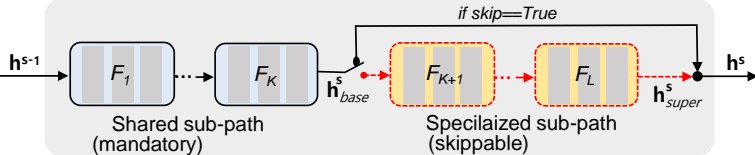

Figure 2: Illustration of a residual stage with a specialized sub-path. A residual stage is split into two sub-paths. While the first (blue) sub-path is mandatory for all sub-networks, the last (yellow) sub-path can be skipped. During training, the last sub-path is encouraged to refine input feature representation $\mathbf{h}_{base}^s$, while minimally changing its distribution. When the specialized sub-path is skipped, a different set of batch normalization operators, called *skip-aware BN*s, are used in the shared sub-path. The shortcuts within residual blocks are not shown for brevity.

Then, what do the residual functions in $\mathbf{F}_{skippable}$ learn during training? This can be further investigated through Taylor expansion (Jastrzebski et al., 2018). Any loss function $\mathcal{L}$ used for training can be approximated with Taylor expansion as follows:

$$\mathcal{L}(\mathbf{h}_{super}^s) = \mathcal{L}(\mathbf{h}_{base}^s + F_{K+1}(\mathbf{h}_K^s) + ... + F_{L-1}(\mathbf{h}_{L-2}^s) + F_L(\mathbf{h}_{L-1}^s)) \tag{4}$$

$$\approx \mathcal{L}(\mathbf{h}_{base}^s + F_{K+1}(\mathbf{h}_K^s) + ... + F_{L-1}(\mathbf{h}_{L-2}^s)) + F_L(\mathbf{h}_{L-1}^s) \cdot \frac{\partial \mathcal{L}(\mathbf{h}_{L-1}^s)}{\partial \mathbf{h}_{L-1}^s} + \mathcal{O}(\cdot) \tag{5}$$

$$...$$

$$\approx \mathcal{L}(\mathbf{h}_{base}^s) + \sum_{j=K+1}^{L} F_j(\mathbf{h}_{j-1}^s) \cdot \frac{\partial \mathcal{L}(\mathbf{h}_{j-1}^s)}{\partial \mathbf{h}_{j-1}^s} \tag{6}$$

In Equations 4 -6, the loss function is iteratively expanded around $\mathbf{h}_{j-1}^s$ $(j = K + 1, ..., L)$. In Equation 5, note that only the first order term $F(\mathbf{h}) \cdot \partial \mathcal{L}(\mathbf{h})/\partial \mathbf{h}$ is left and all high order terms, such as $F^2(\mathbf{h}) \cdot \partial^2 \mathcal{L}(\mathbf{h})/\partial \mathbf{h}^2$, are absored in $\mathcal{O}(\cdot)$. These high order terms in $\mathcal{O}(\cdot)$ can be safely ignored in final Equation 6 since $F$s in $\mathbf{F}_{skippable}$ have small magnitudes, as shown in Equation 3. Thus, in Equation 6, minimizing the loss $\mathcal{L}$ during training optimizes $F_j$ $(j = K + 1, ..., L)$ in the half space of $-\frac{\partial \mathcal{L}(\mathbf{h}_{j-1}^s)}{\partial \mathbf{h}_{j-1}^s}$ to minimize the dot product between $F_j$ and $\frac{\partial \mathcal{L}(\mathbf{h}_{j-1}^s)}{\partial \mathbf{h}_{j-1}^s}$. This implies that every residual function in $\mathbf{F}_{skippable}$ is optimized to learn a function that has a similar effect to gradient descent:

$$F_j(\mathbf{h}_{j-1}^s) \simeq - \frac{\partial \mathcal{L}(\mathbf{h}_{j-1}^s)}{\partial \mathbf{h}_{j-1}^s} \ (j = K + 1, ..., L) \tag{7}$$

Considering these results in Equations 2 and 7 together, we can conjecture that residual functions in $\mathbf{F}_{skippable}$ (1) minimally change the distribution of input features $\mathbf{h}_{base}^s$, but (2) refine them iteratively to minimize the loss for better inference accuracy. In other words, *the skippable sub-path is specialized for refining input features for better inference accuracy while preserving its feature level*. Therefore, $\mathbf{F}_{skippable}$, or the specialized residual blocks, can be skipped without causing much changes of the feature distribution to the next residual stages (Sections 4.3 and 4.4).

### 3.3 TRAINING ADAPTIVE DEPTH NETWORKS

To enforce Equation 2, we propose a training method shown in Algorithm 1, in which two sub-networks, or the *super-net* and the *base-net*, are exploited in a single training loop for sub-paths specialization. The *super-net* and the *base-net* are the largest and the smallest sub-networks, respectively, of the proposed adaptive depth network $M$. For example, the base-net skips every specialized sub-paths in $M$. In contrast, the super-net executes all residual blocks of $M$ without skipping.

In steps 5 and 6, the forward passes of the super-net and the base-net are executed for the same input $x$. The second argument, *skip*, of the model $M$ indicates in which residual stage its specialized sub-path is skipped. In ResNets with 4 residual stages, $16(= 2^4)$ sub-networks become available by varying the *skip* argument. For example, we choose the base-net of a ResNet by setting the second argument to *skip=[True, True, True, True]*. During the training, a wrapper function is used to obtain intermediate features from every residual stage as well as the outputs of the sub-networks. For instance, during the forward pass of the base-net, intermediate features from $N_s$ residual stages, or $\mathbf{h}_{base} = [\mathbf{h}_{base}^1, ..., \mathbf{h}_{base}^{N_s}]$, and output $\hat{y}_{base}$ are obtained.

---

**Algorithm 1** Training with sub-paths specialization. $N_s$ is the number of residual stages.

---

1: Initialize an adaptive depth network $M$
2: **for** i=1,...,$n_{iters}$ **do**
3:    Get next mini-batch of data $x$ and label $y$
4:    $optimizer.zero\_grad()$
5:    $[\hat{y}_{super}, \mathbf{h}_{super}] = M(x, skip=[False, ... , False])$     ▷ Forward pass for the super-net of $M$
6:    $[\hat{y}_{base}, \mathbf{h}_{base}] = M(x, skip=[True, ... , True])$     ▷ Forward pass for the base-net of $M$
7:    $loss = \alpha\, criterion(y, \hat{y}_{super}) + (1-\alpha)\{\sum_{s=1}^{Ns} D_{KL}(\mathbf{h}_{super}^s \| \mathbf{h}_{base}^s) + D_{KL}(\hat{y}_{super} \| \hat{y}_{base})\}$
8:    $loss.backward()$
9:    $optimizer.step()$
10: **end for**

---

In step 7, we modify the loss by adding Kullback Leibler (KL) divergence between $\mathbf{h}_{base}$ and $\mathbf{h}_{super}$, given in Equation 2, as an regularization term:

$$loss = \alpha\, criterion(y, \hat{y}_{super}) + (1-\alpha)\{\sum_{s=1}^{Ns} D_{KL}(\mathbf{h}_{super}^s \| \mathbf{h}_{base}^s) + D_{KL}(\hat{y}_{super} \| \hat{y}_{base})\} \quad (8)$$

$\hat{y}_{super}$ and $\hat{y}_{base}$ are included in the regularization term since they are last features from the sub-networks. The hyperparameter $\alpha$ controls the strength of the regularization term.

In previous anytime networks, all sub-networks are explicitly trained either jointly or sequentially (Yu & Huang, 2019b), and, hence, the total training time is proportional to the number of supported sub-networks. In contrast, our approach does not train all sub-networks jointly or sequentially. Instead, in Algorithm 1, only two sub-networks are exploited during training for sub-paths specialization, and, hence, the total training time is no greater than training two sub-networks individually. However, at test time, the specialized sub-paths can be exploited selectively to construct $2^{N_s}$ sub-networks of various depths. We demonstrate this result in Section 4.3.

### 3.4 Skip-Aware Batch Normalization for Shared Sub-paths

In our adaptive depth networks, each residual stage expects the distribution of input features is not affected by the choices of internal paths of its previous residual stages. However, the shared sub-paths inside each residual stage still need adaptability to handle different sub-networks. Originally, batch normalization (BN) (Ioffe & Szegedy, 2015) was proposed to handle internal covariate shift during training non-adaptive networks by normalizing features. In our adaptive depth networks, however, internal covariate shifts occur in shared sub-paths when different sub-networks are used. To handle this internal covariate shifts, switchable BN operators, called *skip-aware BNs*, are used in shared sub-paths. For example, at each residual stage, different BN operators are used by the layers in the shared sub-path when specialized sub-path is skipped. The effectiveness of switchable BNs has been demonstrated in neural networks with adaptive widths (Yu & Huang, 2019a) and adaptive resolutions (Zhu et al., 2021), and we apply them for our adaptive depth networks. The amount of parameters for skip-aware BNs is negligible. In ResNet50, skip-aware BNs increase the parameters by 0.07%.

## 4 Experiments

To demonstrate the effectiveness of our approach, we conduct experiments on ImageNet classification and COCO object detection and instance segmentation benchmarks. Three representative residual networks both from CNNs and vision transformers are chosen as base models; MobileNet V2 (Sandler et al., 2018) is a lightweight CNN model, ResNet (He et al., 2016) is a larger CNN model, and ViT (Dosovitskiy et al., 2021) is a representative vision transformer. For CNN models, every residual stage is split equally into two sub-paths and its second sub-path is skippable. Since the vision transformer (ViT-b) has no residual stages, 12 consecutive encoder blocks are divided into 4 groups, resembling CNN models, and the last encoder block of each group is used as a skippable sub-path. Since vision transformers exploit layer normalization instead of batch normalization (Yao et al., 2021), we apply switchable layer normalization operators in shared sub-paths instead of switchable BNs. During training, every skippable sub-path is specialized according to Algorithm 1. We use *'-ADN'* to

denote our adaptive depth networks. Since our adaptive depth networks have many parameter-sharing sub-networks in a single model, we indicate which sub-network is used for experiments in parenthesis, e.g., -ADN (super-net). For our sub-networks, boolean values in the parenthesis, e.g., (TTFF), are also used to indicate in which residual stages the sub-path is skipped. For example, -ADN (base-net) is equivalent to -ADN (TTTT).

## 4.1 IMAGENET CLASSIFICATION

We evaluate our method on ILSVRC2012 dataset (Russakovsky et al., 2015) that has 1000 classes. The dataset consists of 1.28M training and 50K validation images. For CNN models, we follow most training settings in the original papers (He et al., 2016; Sandler et al., 2018), except that ResNet models are trained for 150 epochs. For ViT-b, we follow DeIT's training recipe (Touvron et al., 2020; pyt, 2022). The hyperparameter $\alpha$ is set to $0.5$ for all models. For fair comparison, individual networks are trained in the same training settings of their corresponding adaptive depth networks.

Table 1: Results on ImageNet. Models with '-Base' have the same depth as the base-net of corresponding adaptive depth network. Latency is measured on an RTX 3090 device (batch size = 64).

| Baseline | Model | Params | FLOPs | Acc@1 | Acc@5 | Latency |
|---|---|---|---|---|---|---|
| ResNet50 | **ResNet50-ADN (super-net)** | 25.58M | 4.11G | 77.6% | 93.7% | 43.2ms |
| | **ResNet50-ADN (base-net)** | | 2.58G | 76.1% | 93.2% | 27.6ms |
| | ResNet50 (individual network) | 25.56M | 4.11G | 76.7% | 93.2% | 43.2ms |
| | ResNet50-Base (individual network) | 17.11M | 2.58G | 75.0% | 92.3% | 27.6ms |
| | Early-exiting branches | | 4.11G | 75.2% | - | 43.2ms |
| | (Zhang et al., 2019; 2022) | 28.82M | 3.52G | 72.8% | - | 39.0ms |
| | (Li et al., 2019) | | 2.05G | 58.5% | - | 27.4ms |
| | Slimmable widths | | 4.1G | 76.0% | - | 43.2ms |
| | 1.0x, 0.75x, 0.5x widths | 25.6M | 2.0G | 75.6% | - | 33.8ms |
| | (Yu et al., 2018; Yu & Huang, 2019a) | | 1.0G | 74.0% | - | 21.2ms |
| MobileNet V2 | **MV2-ADN (super-net)** | 3.72M | 0.32G | 72.7% | 90.8% | 15.5ms |
| | **MV2-ADN (base-net)** | | 0.25G | 70.7% | 89.8% | 13.2ms |
| | MV2 (individual network) | 3.50M | 0.32G | 72.1% | 90.3% | 15.5ms |
| | MV2-Base (individual network) | 2.99M | 0.25G | 70.7% | 89.6% | 13.2ms |
| | MutualNet (Yang et al., 2020b) | - | 0.32G | 72.9% | - | - |
| | AlphaNet-0.75x (Wang et al., 2021a) | - | 0.21G | 70.5% | - | - |
| ViT-b/32 | **ViT-b/32-ADN (super-net)** | 88.25M | 2.95G | 76.4% | 92.7% | 39.5ms |
| | **ViT-b/32-ADN (base-net)** | | 2.00G | 73.2% | 91.4% | 26.7ms |
| | ViT-b/32 (individual network) | 88.22M | 2.95G | 75.9 % | 92.5% | 39.5ms |
| | ViT-b/32-Base (individual network) | 69.33M | 2.00G | 73.1% | 90.8% | 26.7ms |

The results in Table 1 show that our adaptive depth networks consistently outperform individual counterpart networks even though many sub-networks share parameters in a single network. We conjecture that these improvements result from distilling the knowledge from $\mathbf{h}^s_{super}$ to $\mathbf{h}^s_{base}$ at each residual stage and the iterative refinement at the skippable sub-paths, shown in Equation 7. In Table 1, ResNet50-ADN is also compared with two representative anytime networks (Zhang et al., 2019; Yu & Huang, 2019a) that can control the depths and the widths, respectively. Our adaptive depth networks achieve better performance than these competitors. In particular, unlike competitors, the reduction of FLOPs of ResNet-ADN is closely translated into actual acceleration. For instance, when the FLOPs of ResNet50-ADN (base-net) is reduced by 37.3%, its inference latency is also similarly reduced by 36.2%. In contrast, although S-ResNet50 (0.75x) requires 22% less FLOPs than ResNet50-ADN (base-net), it shows 22.4% longer inference latency in practice. Since S-ResNet50's network depth is not changed by reducing widths, it still needs to perform similar number of GPU kernel invocations in practice. In early-exiting anytime networks, effective depths of network can be reduced. But as shown in Zhang et al. (2019)'s work, sub-networks of early-exiting networks manifest significantly low inference accuracy since they only exploit low-level features. In MobileNetV2, our MV2-ADN is compared with recent adaptive networks that control network widths and resolutions. While these networks exploit recent training techniques such as AutoAugment (Cubuk et al., 2019) and input

resolution variations (Wang et al., 2021a), our MV2-ADN achieves similar performance without such bells and whistles. Finally, our result with ViT-b/32-ADN demonstrates that our approach is generally applicable to residual networks including vision transformers.

Table 2: Comparison of state-of-the-art efficient inference methods for ResNet 50 on ImageNet. † denotes static pruning methods, ∗ denotes new filter design, ⋆ denotes dynamic networks.

| Model | Params | FLOPs | ↓FLOPs | Acc@1 | Acc@5 |
|---|---|---|---|---|---|
| PFP-A-ResNet50 (Liebenwein et al., 2020)† | 20.9M | 3.7G | 10% | 75.9% | 92.8% |
| Versatile-ResNet50 (Wang et al., 2018b)∗ | 11.0M | 3.0G | 27% | 74.5% | 91.8% |
| Dynamic Slimmable (Li et al., 2021)⋆ | - | 3.1G | 24% | 76.6% | - |
| **ResNet50-ADN (TTFF)** | 25.6M | 3.4G | 17% | 77.0% | 93.4% |
| GReg-1 (Wang et al., 2021b)† | - | 2.8G | 33% | 76.1% | - |
| Ghost-ResNet50-2 (Han et al., 2020)∗ | 13.0M | 2.2G | 46% | 75.0% | 92.3% |
| DR-ResNet50 ($\alpha$=2.0) (Zhu et al., 2021)⋆ | 30.5M | 2.3G | 44% | 75.3% | 92.2% |
| **ResNet50-ADN (base-net)** | 25.6M | 2.6G | 37% | 76.1% | 93.2% |

In Table 2, two sub-networks of ResNet50-ADN are compared with several state-of-the-art efficient inference methods. Both sub-networks of ResNet50-ADN achieve very competitive results compared to many state-of-the-art methods. Some of these state-of-the-art methods require unusual training techniques to provide accuracy-efficiency trade-offs. For example, in dynamic slimmable networks (Li et al., 2021), 4 sub-networks are randomly sampled at every batch iteration until all sub-networks are converged. In contrast, our approach exploits only two sub-networks, or super-net and base-net, during training according to Algorithm 1. Further, unlike static pruning methods, since the sub-networks of our approach share parameters in a single model, they can adapt their depths instantly at runtime. The result with ResNet50-ADN (base-net) also demonstrates that non-skippable sub-paths of our adaptive depth networks learn compact feature representations effectively while skippable sub-paths are enforced to preserve the level of input features during training.

## 4.2 ABLATION STUDY

Table 3: Ablation analysis with ResNet50-ADN. Applied components are checked with $\sqrt{}$ mark.

| Sub-paths specialization | Skip-aware BNs | Acc@1 | |
|---|---|---|---|
| | | Super-net | Base-net |
| | | 75.2% (↓ 1.5%) | 72.2% (↓ 2.8%) |
| $\sqrt{}$ | | 76.1% (↓ 0.6%) | 74.9 % (↓ 0.1%) |
| | $\sqrt{}$ | 76.6% (↓ 0.1%) | 75.1% (↑ 0.1%) |
| $\sqrt{}$ | $\sqrt{}$ | 77.6% (↑ 0.9%) | 76.1% (↑ 1.1%) |

We conduct ablation study on ImageNet classification to investigate the influence of two key components of the proposed adaptive depth networks: (1) sub-paths specialization and (2) skip-aware BNs. When our sub-path specialization is not applied, the loss in Algorithm 1 is modified to $loss = \frac{1}{2}\{criterion(\boldsymbol{y}, \boldsymbol{\hat{y}}_{super}) + criterion(\boldsymbol{y}, \boldsymbol{\hat{y}}_{base})\}$. The results are shown in Table 3. When neither of them is applied, the inference accuracy of the super-net and the base-net is significantly lower than that of individual networks by 1.5% and 2.8%, respectively. This result shows the difficulty of training multiple sub-networks in anytime networks. During training, uncoordinated training signals from the super-net and the base-net decrease the effect of training, resulting in worse performance than individual networks. When one of the two components is applied individually, the performance is still slightly worse than individual networks'. Finally, when both sub-paths specialization and skip-aware BNs are applied together, ResNet50-ADN achieves significantly better performance than individual networks, both in the super-net and the base-net. This result demonstrates that both sub-paths specialization and skip-aware BNs are essential for the proposed adaptive depth networks.

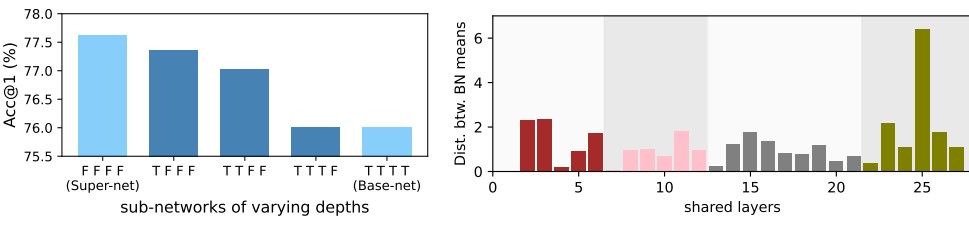

(a) Acc@1 of various sub-networks       (b) Distances btw. skip-aware BNs in shared sub-paths

Figure 3: **(a)** Performance while varying the depth of ResNet50-ADN. The labels under each bar are boolean values indicating the residual stages that skip their specialized sub-path. **(b)** The distances between the means of skip-aware BNs in the shared sub-paths of ResNet50-ADN. Bars with different colors belong to different residual stages.

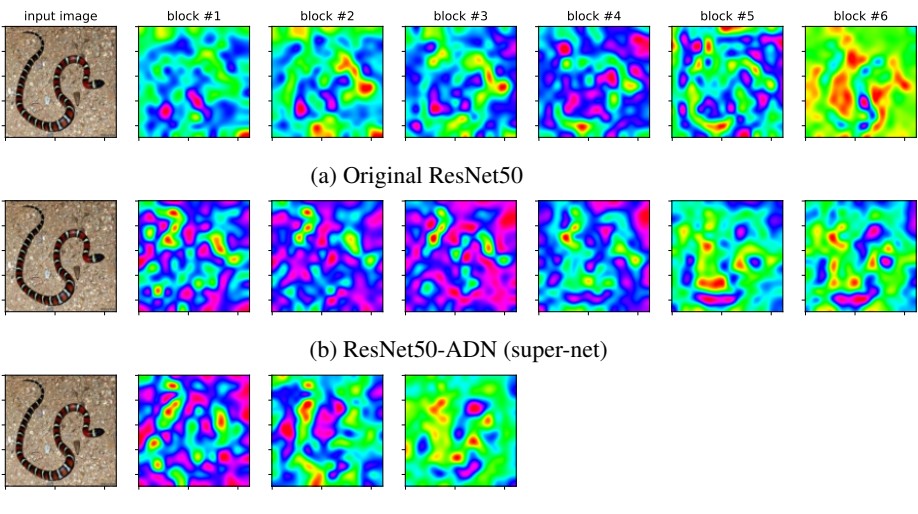

(a) Original ResNet50

(b) ResNet50-ADN (super-net)

(c) ResNet50-ADN (base-net)

Figure 4: Class Activation Map images of the 3rd stages of ResNet50s. **(a)** Original ResNet50's activation regions change gradually across all blocks, indicating that every block performs similar functions. **(b)** In ResNet50-ADN (super), the non-skippable first 3 blocks have extensive hot activation regions, implying active learning of new level features. In contrast, the skippable last 3 blocks have far less activation regions and they are gradually refined around the target. **(c)** The base-net's final activation map is very similar to the super-net's, but has less hot activations around the target object.

### 4.3 VARYING NETWORK DEPTHS AT TEST TIME

One important advantage of sub-paths specialization is that each residual stage's choice of internal paths does not affect the following residual stages. Therefore, each residual stage can independently make a decision about skipping its specialized sub-path. This implies that our adaptive depth networks can support $2^{N_s}$ sub-networks at test time, when $N_s$ residual stages are given. Figure 3-(a) shows the performance of ResNet50-ADN when its depth is varied at test time. Among them, only the super-net and the base-net are used for training in Algorithm 1. The other sub-networks are built by applying different combinations of skipping decisions at test time. Although these sub-networks are not trained explicitly, their performance does not drop significantly. Instead, they show gradual degradation of performance as the depth of sub-networks becomes gradually shallower.

### 4.4 ANALYSIS OF SUB-PATHS SPECIALIZATION

The premise of sub-paths specialization is that the feature distribution from each residual stage is not much affected whether its specialized sub-path is skipped or not; i.e, $D_{KL}(\mathbf{h}_{base}^s || \mathbf{h}_{super}^s) \rightarrow 0$. We can verify this by inspecting the input distribution of the next residual stages. Figure 3-(b) shows the

distances between the means of switchable BNs in the shared sub-paths of ResNet50-ADN. In Figure 3-(b), the distance at every residual stage's first layer is close to zero. This demonstrates that their previous residual stages produce feature distributions of the same means regardless of whether the specialized sub-path is skipped or not. Figure 3-(b) also shows that the shared layers except the first layers of each residual stage have very different input distributions, demonstrating the necessity of skip-aware BNs in the shared sub-paths.

To further verify that the non-skippable sub-paths and the skippable sub-paths, respectively, focus more on learning new level features and refining learned features, we visualize the activation of 3rd residual stage of ResNet50-ADN using Grad-CAM (Selvaraju et al., 2017) in Figure 4. The 3rd residual stage of ResNet50-ADN has six residual blocks and the last three blocks are skippable. In Figure 4-(a), the activation regions of original ResNet50 changes gradually across all consecutive blocks. This implies that all blocks are involved in gradual learning of new level features as well as the refinement of the learned features. In contrast, in Figure 4-(b), ResNet50-ADN (super-net) manifests very different progressions of activation regions in two sub-paths. In the first three residual blocks, we can observe lots of hot activation regions in wide areas, suggesting active learning of new level features. In contrast, significantly less activation regions are found in the skippable last three blocks and they are gradually concentrated around the target object, demonstrating the refinement of learned features. Further, in Figure 4-(c), we can observe that the final activation map of the ResNet50 (base-net) is very similar to the super-net's final activation map in (b), implying that they are at the same feature level as suggested in Equation 2. However, the final activation map of the base-net has less hot activation regions around the target object than super-net's, potentially resulting in lower inference accuracy than super-net's more refined features.

### 4.5 OBJECT DETECTION AND INSTANCE SEGMENTATION

Table 4: Object detection and instance segmentation results on MS COCO dataset.

| Detector | Backbone | FLOPs | Individual Networks | | ResNet-ADN (ours) | |
|---|---|---|---|---|---|---|
| | | | Box AP | Mask AP | **Box AP** | **Mask AP** |
| Faster-RCNN | ResNet50 | 207.07G | 36.4 | | 37.8 | |
| (Ren et al., 2017) | ResNet50-Base | 175.66G | 32.4 | | 34.0 | |
| Mask-RCNN | ResNet50 | 260.14G | 37.2 | 34.1 | 38.3 | 34.1 |
| (He et al., 2017) | ResNet50-Base | 228.73G | 32.7 | 29.9 | 34.1 | 31.2 |

In order to investigate the generalization ability of our approach, we use MS COCO 2017 datasets on object detection and instance segmentation tasks using representative detectors. We compare individual ResNets and our adaptive depth ResNets-ADN as backbone networks of the detectors. For training of detectors, we use Algorithm 1 with slight adaptation. For object detection, the intermediate features $\mathbf{h}^s_{base}$ and $\mathbf{h}^s_{super}(s = 1..N_s)$ can be obtained directly from backbone network's feature pyramid networks (FPN) (Lin et al., 2017), and, hence, a wrapper function is not required to extract intermediate features. All networks are trained on `train2017` for 12 epochs from ImageNet pretrained weights, following the training settings suggested in (Lin et al., 2017). Table 4 shows the results on `val2017` containing 5000 images. Our adaptive depth backbone networks significantly outperform individual static backbone networks in terms of COCO's standard metric AP.

## 5 CONCLUSIONS

We present a novel approach to adaptive depth networks that is generally applicable to residual networks, including both convolution networks and vision transformers. Unlike previous approaches, our adaptive depth networks do not require auxiliary add-on networks or classifiers to control network depths. Instead, our approach train sub-paths of a single network to have a special property, with which the sub-paths can be skipped for efficiency at marginal loss of prediction accuracy. We discuss the theoretic rationale behind this sub-paths specialization, and present a simple and practical training method. While our training method uses only two sub-networks, it makes various sub-networks with different depths available at test time. Through extensive experiments with both convolution networks and vision transformers, we demonstrate that the proposed adaptive depth networks outperform individual static networks while achieving actual inference speed-up.

## 6 REPRODUCIBILITY

For reproducibility, we provide source codes and the links to the pretrained weights as the supplementary material. Code and models will be available in authors' public repository. All programs for training and evaluation run on PCs equipped with two or four Nvidia RTX 3090 GPUs and an Intel i9-10900X CPU @3.7GHz.

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
