# OpenReview forum: "Specialization of Sub-paths for Adaptive Depth Networks"
_ICLR.cc/2023/Conference — Submitted to ICLR 2023_

### Official Review · Reviewer_qCMH · 2022-10-22

**Confidence:** 4
**Correctness:** 2
**Technical Novelty And Significance:** 2
**Empirical Novelty And Significance:** 2
**Recommendation:** 5

**Clarity, Quality, Novelty And Reproducibility:**

- The presentation is fairly clear, apart from the technical aspects discussed above.
- The quality is sufficient, but suffers primarily due to the lack of careful notation and rigour in experimentation.
- There is some novelty in skipping the computation in the residual blocks as proposed here. However, it could still be considered as a particular implementation of model pruning, which is a well-studied strategy.
- The code is provided for review. There is some discrepancy w.r.t. the paper description, hence I'd assume the work could be diffuclt to reproduce exactly without the original code.

**Strength And Weaknesses:**

Strengths
- Parameter redundancy in neural networks is well known empirically, and it is certainly a natural idea to exploit it by learning to skip groups of subsequent layers.
- The implementation of the approach seems sufficiently simple.
- There is some effort in providing formal justification to motivate the approach (Eq. 2-5).
- The scope of the experiments is good and encompasses classification, instance segmentation and object detection in addition to the ablation study.

Weaknesses

- Overall, the demonstrated empirical improvements are very incremental. For example, on image classification they are somewhat comparable to Wang et al. 2021 (c.f. Tab. 2).
- The baseline does not seem sufficiently strong, in fact 2-3% lower than that reported in the original work (He et al., 2015). A short training schedule may actually disadvantage the baseline, since it has a larger number of parameters. (For the proposed method the convergence may happen faster, since due to the KL-divergence the number of training parameters is implicitly reduced).
- Isn’t the “super-net” the exact copy of the baseline, but with additional KL-divergence term? Why would the baseline have lower classification accuracy?
- There is generally lack of technical rigour which leaves some room for ambiguity and impedes the flow of the presentation. For example, Eq. 2-5 equate the expectation of an n-dimensional vector with a scalar value, which does not seem meaningful. Moreover, the implementation of these constraints (if one follows them somewhat intuitively) is specifically with the KL-divergence. I believe the equations should reflect that. Another example is Eq. (5): is it assumed to follow from Eq. (4), or is it a standalone assumption?

Somewhat minor:
- Inconsistent notation, e.g. h_super in Eq. 1 (with or without parenthesis?); same for h_base.
- Eq. 7 seems sloppy (what is in O(.), partial derivative vs. total derivative).
- the special treatment of the 23 blocks in ResNet-101 seems to be rather contrived.
- I do not see much support in the KL-divergence to "preserve the semantic level of input features".

Further notes:
- Table 1: Please provide details of the evaluation protocol, e.g. what data and criterion was used to select the best model for evaluation?
- Fig 4c does not appear very convincing: the features learned by the base network are somewhat reminiscent of the “super-net”.
- Table 3 unclear: if neither of the two components are used, this should be the baseline setting, shouldn’t it?
- The implementation seems to use a hyperparater T set to 4 in the KL-divergence terms. It is not mentioned in the paper. What is its use?

**Summary Of The Paper:**

This work develops a method to reduce computational costs of a deep residual network without substantial detriment to its prediction accuracy. Specifically, it considers the second half of the layers in each residual block as disposable, i.e. one could skip those layers at inference time to reduce the runtime. To demonstrate this, the work experiments with minimising the KL-divergence between the output of a residual block and the features produced only by the first half of the layers. The experiments on image classification, object detection and instance segmentation indicate that network efficiency can be almost doubled only at a moderate loss in the prediction accuracy.

**Summary Of The Review:**

This work achieves reduction in model efficiency with tolerable loss in accuracy. These results are encouraging. However, the experimental protocol requires more elaboration, much like the technical aspects of the presentation.

---

> ### Author Response · Authors · 2022-11-15
> **Response to Reviewer R4 (Part 2)**
>
> **R4-Q4**: Inconsistent notation, e.g. h_super in Eq. 1 (with or without parenthesis?); same for h_base.
>
> **R4-A4**: In the revision, the parenthesis are removed for consistency. Thanks!
>
> **R4-Q5**: Eq. 7 seems sloppy (what is in O(.), partial derivative vs. total derivative).
>
> **R4-A5**: The high order term O(.) absorbs 2nd, 3rd, and more high order 'partial' derivative terms. In the revision, we update the discussion following Equation (4)-(6) to make this clear.
>
> **R4-Q6**: the special treatment of the 23 blocks in ResNet-101 seems to be rather contrived.
>
> **R4-A6**: We agree! Experiment with ResNet-101 seems not much necessary. Instead, many reviewers asked experiments with more recent vision transformers to show the generality and novelty of our work. For better use of limited paper space, in the revision, we replace ResNet101 results with ViT-b/32 results in Table 1. In ViT, our approach still achieve better performance than individual ViT. **To the best of authors' knowledge, our work is the first general approach to adaptive networks demonstrating consistent performance improvements for both CNNs and vision transformers.** Please see the following results with ViT in Table 1.
> model | FLOPS |Acc@1 | Acc@5 | latency
> ---- | ----  | ----- | ------| ----
> **ViT-b/32-ADN (super-net)** |  2.95G | 76.4\%  |92.7\% | 39.5ms
> **ViT-b/32-ADN (base-net)** |  2.00G | 73.2\% | 91.4\% | 26.7ms
> ViT-b/32 (individual network) |2.95G | 75.9\% | 92.5\% | 39.5ms
> ViT-b/32-Base (individual network) | 2.00G | 73.1\% | 90.8\% | 26.7ms
>
> **R4-Q7**: I do not see much support in the KL-divergence to "preserve the semantic level of input features"
>
> **R4-A7**: Thanks for this suggestion! In the revision, we update Equation (2) to clarify that we use KL-divergence to minimize the *gap between the distributions* of $h_{base}$ and
> $h_{super}$. We update many sentences to "the *level (or distribution)* of input features" instead of "the semantic level of input features".
>
> **R4-Q8**: Table 1: Please provide details of the evaluation protocol, e.g. what data and criterion was used to select the best model for evaluation?
>
> **R4-A8**: We update the first paragraph of Section 4 to include the following: "We use three representative residual networks both from CNNs and vision transformers as base models; MobileNet V2 (Sandler et al., 2018) is a lightweight CNN model, ResNet (He et al., 2016) is a larger CNN model, and ViT (Dosovitskiy et al., 2021) is a representative vision transformer."
>
> **R4-Q9**: Fig 4c does not appear very convincing: the features learned by the base network are somewhat reminiscent of the “super-net”.
>
> **R4-A9**: Since both 'super-net' and 'base-net' are optimized to produce features with similar distributions, we belive, this result is very expected. This result is reproducable with any GradCAM programs.
>
> **R4-Q10**: Table 3 unclear: if neither of the two components are used, this should be the baseline setting, shouldn’t it?
>
> **R4-A10**: When neither of the two components are used, we still use two forward passes followed by one backward pass, shown in Algorithm 1. But, the loss function is modified as follows:
>
> $loss= \frac{1}{2} \\{criterion(y, \hat{y}\_{super}) + criterion(y, \hat{y}\_{base}) \\}$.
>
> In the revision, we update Section 4.2 to clarify this.
>
> **R4-Q11**: The implementation seems to use a hyperparater T set to 4 in the KL-divergence terms. It is not mentioned in the paper. What is its use?
>
> **R4-A11**: Hyperparameter T is 'temperature' when KL divergence is used for distillation. Since T=4 is typically used in knowledge distillation as shown in [1], we use T=4 without hyperparameter tuning.
>
> [1] Cho et al. On the Efficacy of Knowledge Distillation, CVPR 2019

---

> ### Author Response · Authors · 2022-11-15
> **Response to Reviewer R4 (Part 1)**
>
> Thanks to reviewer R4 for helping improve our paper! We address the concerns as follows:
>
> **R4-Q1**: Overall, the demonstrated empirical improvements are very incremental. For example, on image classification they are somewhat comparable to Wang et al. 2021 (c.f. Tab. 2).
>
> **R4-A1**: In Table 2, we compare our sub-networks with state-of-the-art efficient inference methods. They include (1) static pruning methods, (2) new filter design, and (3) dynamic networks. These networks are either static networks or do not give users the freedom to control networks' resource consumption. For instance Wang et al. (2021)'s work is a static pruning method. Unlikely, our sub-networks share parameters in a single model and can switch to different sub-networks at runtime. Nevertheless, our sub-networks still achieve better or similar performance. To make this clear, in the revision, we classify competing efficient inference methods and annotate them in Table (2).
>
> **R4-Q2**: The baseline does not seem sufficiently strong, in fact 2-3% lower than that reported in the original work (He et al., 2015). A short training schedule may actually disadvantage the baseline, since it has a larger number of parameters. (For the proposed method the convergence may happen faster, since due to the KL-divergence the number of training parameters is implicitly reduced).
>
> **R4-A2**: ResNet's original work (He et al., 2015) reports 10-crop performance. In other words, they make 10 separate predictions and take an average of them. This performance method was used in ImageNet competition, but most later papers report single-crop performance. For example, original work reports 78.47% top-1 accuracy (10-crop Acc@1), but most papers use pytorch's 76.13% single-crop performance as their baseline. In our work, for fair comparison, our adaptive networks and counterpart individual networks are trained in the same training settings. And, our ResNet50 (individual networks) shows 76.7% top-1 accuracy, which is slightly higher than pytorch's. In the revision, we update the first paragraph of Section 4.1 to clarify that we apply same training settings for fair comparison.
>
> **R4-Q3**: Isn’t the “super-net” the exact copy of the baseline, but with additional KL-divergence term? Why would the baseline have lower classification accuracy? There is generally lack of technical rigour which leaves some room for ambiguity and impedes the flow of the presentation. For example, Eq. 2-5 equate the expectation of an n-dimensional vector with a scalar value, which does not seem meaningful. Moreover, the implementation of these constraints (if one follows them somewhat intuitively) is specifically with the KL-divergence. I believe the equations should reflect that. Another example is Eq. (5): is it assumed to follow from Eq. (4), or is it a standalone assumption?
>
> **R4-A3**: We appreciate this comments and suggestions! We have updated theoretic parts in Section 3.2 extensively. In particular, Equations (2)-(8) in the original submission were redundant and caused confusion since they did not match exactly
> the implementation in Algorithm (1).
> As suggested by reviewers, we have updated these equations.
> The most important one is Equation (2) that is updated to:
>
> $D_{KL} (\textbf{h}^s_{super} || \textbf{h}^s_{base}) \to 0$
>
> This Equation matches our implementation and show
> more clearly why our approach improves the performance
> while many sub-networks share parameters in a single model.
> By having Equation (2) as a regularization term in the loss function,
> as shown in Algorithm (1),
> we can obtain distillation effect of transferring the knowledge from
> $\textbf{h}^s_{super}$ to $\textbf{h}^s_{base}$.
> Therefore, we can conjecture that,
> at each residual stage, $\textbf{h}^s_{base}$ learns
> more compact representation from $\textbf{h}^s_{super}$.

---

### Official Review · Reviewer_zGdd · 2022-10-25

**Confidence:** 4
**Correctness:** 3
**Technical Novelty And Significance:** 2
**Empirical Novelty And Significance:** 2
**Recommendation:** 3

**Clarity, Quality, Novelty And Reproducibility:**

- **Clarity:** The paper is clearly written and easy to follow. There are some grammar errors (like “Slimmable widhts” in Table 1) that should be further corrected.

- **Quality.**  The quality is relatively low. The theoretical analysis lacks insight. The experiments are only performed with ResNet and MobileNet, which is not very convincing.

- **Novelty.** The novelty of the proposed method is low. I think it is a simplified version of AlphaNet.

- **Reproducibility.** Code is attached in this submission.



**Strength And Weaknesses:**

Strengths:

- The discussion on the two primary functions of the residual blocks is interesting.
- The paper is well-organized and easy to follow.

Weaknesses:
- The novelty of this paper is questioned. The overall design highly resembles previous work like AlphaNet[1]. The base network in this paper is actually a simplified version of the sampled sub-network in AlphaNet. Besides, while AlphaNet considers multiple scaling dimensions including network depth and width, this paper only considers network depth. As for the training method, both the AlphaNet and this paper adopt KL divergence to minimize the gap between the sub-network and the super-network.

- Some of the theoretical parts lack insight and are somewhat redundant. For example, by some simple substitution and simplification, one can easily show that Equation (2)-(4) are duplicated. Besides, Equation (6)-(8) are just trivial Taylor expansion of the loss function, which is not strongly related to the proposed method.

- The experiments are not sufficient. According to the paper, the method can be applied to any architectures that are built with residual blocks, including the recent prevailing Vision Transformers [2]. I highly recommend that the authors should perform more experiments using ViT as their baseline to show the effectiveness of the proposed method.


[1] Wang, Dilin, et al. "AlphaNet: improved training of supernets with alpha-divergence." International Conference on Machine Learning. PMLR, 2021

[2] Dosovitskiy, Alexey, et al. "An image is worth 16x16 words: Transformers for image recognition at scale." arXiv preprint arXiv:2010.11929 (2020).



**Summary Of The Paper:**

This paper proposes a method to achieve anytime networks by controlling the network depth during runtime. The method focuses on the typical ResNet-like architectures and proposes to divide the residual blocks in each stage into two parts that are responsible for feature learning and feature refinement, respectively. A new training method is proposed to achieve the above goal, which is implemented as minimizing the difference between the super network and the base network in both final predictions and pooled intermediate features. Experimental results show that the proposed can achieve better trade-offs than the baselines and previous works.


**Summary Of The Review:**

This paper presents a new method to achieve controllable depth during runtime. However, the proposed method highly resembles previous work AlphaNet. Besides, the lack of experiments makes the paper not very convincing. As a result, I lean towards rejecting this paper.

---

> ### Author Response · Authors · 2022-11-15
> **Response to Reviewer R3 (Part 2)**
>
>
> **R3-Q3**: The experiments are not sufficient. According to the paper, the method can be applied to any architectures that are built with residual blocks, including the recent prevailing Vision Transformers [2]. I highly recommend that the authors should perform more experiments using ViT as their baseline to show the effectiveness of the proposed method.
>
> **R3-A3**: We are very grateful for this suggestion! Unlike previous adaptive networks, the proposed sub-paths specialization exploits the property of residual networks, and, hence, it is general to both CNNs and vision transformers (ViTs).
> As suggested by several reviewers, we apply our method to ViT to demonstrate its generality and novelty.
> We are extremely happy to report our evaluation results in Table 1.
> (Evaluation details on ViT can be found in the revised Section 4.)
>
> model | FLOPS |Acc@1 | Acc@5 | latency
> ---- | ----  | ----- | ------| ----
> **ViT-b/32-ADN (super-net)** |  2.95G | 76.4\%  |92.7\% | 39.5ms
> **ViT-b/32-ADN (base-net)** |  2.00G | 73.2\% | 91.4\% | 26.7ms
> ViT-b/32 (individual network) |2.95G | 75.9\% | 92.5\% | 39.5ms
> ViT-b/32-Base (individual network) | 2.00G | 73.1\% | 90.8\% | 26.7ms
>
> Even though we did minimal hyperparameter tuning due to limited computing resources,
> our results with ViT outperform individual networks' performance.
> (We only reduce the batch size and LR  to half in PyTorch's standard training scripts.)
>
> **To the best of authors' knowledge, our work is the first general approach to
> adaptive networks demonstrating consistent performance improvements for both CNNs and vision transformers.**

---

> ### Author Response · Authors · 2022-11-15
> **Response to Review R3 (Part 1)**
>
> Thanks to reviewer R3 for helping improve our paper! We address the concerns as follows:
>
> **R3-Q1**: The novelty of this paper is questioned. The overall design highly resembles previous work like AlphaNet[1]. The base network in this paper is actually a simplified version of the sampled sub-network in AlphaNet. Besides, while AlphaNet considers multiple scaling dimensions including network depth and width, this paper only considers network depth. As for the training method, both the AlphaNet and this paper adopt KL divergence to minimize the gap between the sub-network and the super-network.
>
> **R3-A1**: Thanks for reporting this missed related work. We carefully checked AlphaNet[1] for comparison and include it in Table 1. Although our work shares a general spirit of adaptive networks with AlphaNet, the two papers are very different in many ways:
>
> 1) Although AlphaNet[1] claims considering both network depth and width conceptually (in Figure 1 of [1]), they do not provide a method to build sub-networks in depth-wise and AlphaNet[1] only considers network widths in actual evaluation. In the paper[1], the evaluation is only performed using slimmable networks (mobilenet v1 and v2). **To the best of authors' knowledge, our work is the first approach to adaptive depth networks demonstrating consistent performance improvements**.
> 2) AlphaNet[1] uses KL divergence between the outputs from sub-networks. In our work, we also use KL divergence to minimize the gap between intermediate features at every residual stage of two sub-networks. In the original submission, this was not clearly discussed. In the revision, we update Equation (2), (8), and Algorithm 1 to clarify this.
> 3) AlphaNet[1] follows 'sandwich rule' from universal slimmable networks [3] for training. With sandwich rule, at each training iteration, four sub-networks are sampled, or the largest, the smallest, and two random sub-networks. This means AlphaNet[1] uses all sub-networks during training. In contrast, our work uses only two sub-networks for training, or the super-net and the smallest base-net. In our work, other sub-networks are never explicitly used for training. However, as shown in Figure 3(a), other sub-networks can be used at test time and they provide inference accuracy proportional to their network depths. Therefore, in most adaptive networks, including AlphaNet[1], the total training time is linearly proportional to the number of supported sub-networks. In contrast, in our work, the total training time is no greater than training two sub-networks. This reduced total training time is another important contribution of our work. In the revision, we include above discussion in Introduction and Section 3.3.
>
> [1] Wang, Dilin, et al. "AlphaNet: improved training of supernets with alpha-divergence." International Conference on Machine Learning. PMLR, 2021\
> [2] Dosovitskiy, Alexey, et al. "An image is worth 16x16 words: Transformers for image recognition at scale." arXiv preprint arXiv:2010.11929 (2020).\
> [3] Yu et al., Universally slimmable networks and improved training techniquues, ICCV 2019.
>
> **R3-Q2**: Some of the theoretical parts lack insight and are somewhat redundant. For example, by some simple substitution and simplification, one can easily show that Equation (2)-(4) are duplicated. Besides, Equation (6)-(8) are just trivial Taylor expansion of the loss function, which is not strongly related to the proposed method.
>
> **R3-A2**: Equations (2)-(5) in the original submission were
> redundant and caused confusion since they did not match exactly
> the implementation in Algorithm (1).
> As suggested by reviewers, we have updated these equations.
> The most important one is Equation (2) that is updated to:
>
> $D_{KL} (\textbf{h}^s_{super} || \textbf{h}^s_{base}) \to 0$
>
> This Equation matches our implementation and show
> more clearly why our approach improves the performance
> while many sub-networks share parameters in a single model.
> By having Equation (2) as a regularization term in the loss function,
> as shown in Algorithm (1),
> we can obtain distillation effect of transferring the knowledge from
> $\textbf{h}^s_{super}$ to $\textbf{h}^s_{base}$.
> Therefore, we can conjecture that,
> at each residual stage, $\textbf{h}^s_{base}$ learns
> more compact representation from $\textbf{h}^s_{super}$
>
> Another important addition is Equation (7):
> $F_{j+1}(\textbf{h}^{s}_{j}) \simeq  - \frac{\partial \mathcal{L}(  \textbf{h}^{s}_j ) }{\partial \textbf{h}^{s}_j}$.
> We believe that our original discussion was not clear
> about the implication of Taylor expansion in the sub-paths specialization.
> We add Equation (7) to show more clearly that *residual functions in
> the skippable sub-paths are optimized to learn a function
> that has a similar effect to gradient descent.*
> In other words, the residual functions in the skippable sub-paths
> reduce the loss during inference.

---

### Official Review · Reviewer_X8ta · 2022-10-25

**Confidence:** 4
**Clarity, Quality, Novelty And Reproducibility:** 1) The paper is easy to read, and und…
**Correctness:** 3
**Technical Novelty And Significance:** 3
**Empirical Novelty And Significance:** 2
**Recommendation:** 5

**Strength And Weaknesses:**

Pro:
1) The method proposed by the paper to slim a residual network looks reasonable, it changed from a slimmable channels to slimmable sub-paths. It demonstrates in imagenet coco that a wider subnetwork performs better than deeper but thinner network based on resnet-like basenetworks [resnet50/101, mobilenetv2].

2) It includes several theoretical side evidence for using skip subpaths rather than skip channels. e.g. [Caramazza & Coltheart 2006], that looks convincing for me.


Cons:
1) The baselines compared in the paper looks out dated, therefore is less convince for the generalization of the proposed strategy.
I think the author should also demonstrate the effectiveness over stronger baselines for example:

Vit-like networks, [An Image is Worth 16x16 Words: Transformers for Image Recognition at Scale]
Swin-transformers [Swin Transformer: Hierarchical Vision Transformer using Shifted Windows]
Cvt networks [CvT: Introducing Convolutions to Vision Transformers]

They induces stronger results on all these tasks, it might be better we have baselines for them.



**Summary Of The Paper:**

This paper proposes a novel adaptive depth network by inducing a training strategy without the need of additional intermediate gate/classifier. It use a skip-aware BN, with a cost of 0.07% parameter increasing, but reduces the inference cost and achieve better results comparing with the non-adaptive baselines.

**Summary Of The Review:**

Good motivation and direction of research. Simple adaptive skip-paths strategy is proposed, while stronger experiments are needed

---

> ### Author Response · Authors · 2022-11-15
> **Response to Reviewer R2**
>
> Thanks to reviewer R2 for helping improve our paper! We address the concerns as follows:
>
> **R2-Q1**: The baselines compared in the paper looks out dated, therefore is less convince for the generalization of the proposed strategy. I think the author should also demonstrate the effectiveness over stronger baselines for example: Vit-like networks.
>
> **R2-A1**: We are very grateful for this suggestion!
> Unlike previous adaptive networks, the proposed sub-paths specialization exploits the property of residual networks, and, hence, it is general to both CNNs and vision transformers (ViTs).
> As suggested, we apply our method to ViT to demonstrate its generality and novelty.
> We are extremely happy to report our evaluation results in Table 1.
> (Evaluation details on ViT can be found in the revised Section 4.)
>
> model | FLOPS |Acc@1 | Acc@5 | latency
> ---- | ----  | ----- | ------| ----
> **ViT-b/32-ADN (super-net)** |  2.95G | 76.4\%  |92.7\% | 39.5ms
> **ViT-b/32-ADN (base-net)** |  2.00G | 73.2\% | 91.4\% | 26.7ms
> ViT-b/32 (individual network) |2.95G | 75.9\% | 92.5\% | 39.5ms
> ViT-b/32-Base (individual network) | 2.00G | 73.1\% | 90.8\% | 26.7ms
>
> Even though we did minimal hyperparameter tuning due to limited computing resources,
> our results with ViT outperform individual networks' performance.
> (We only reduce the batch size and LR to half in PyTorch's standard training scripts.)
>
> *To the best of authors' knowledge, our work is the first general approach to
> adaptive networks demonstrating consistent performance improvements for both CNNs and vision transformers.*
>
> Further, in the revision, we compare our work with more recent state-of-the-art methods
> in Table 1 and Table 2. They include:
>
> [1] Wang, Dilin, et al. "AlphaNet: improved training of supernets with alpha-divergence." International Conference on Machine Learning. PMLR, 2021 \
> [2] Yang et al. "Mutualnet: Adaptive convnet via mutual learning form network width and resolution." ECCV, 2020 \
> [3] Li et al. "Dynamic slimmable network." CVPR, 2021
>
> *A number of important changes have been made in this revision.
> Please check the **general response** above and the revised paper.*

---

### Official Review · Reviewer_ipWP · 2022-10-27

**Confidence:** 4
**Correctness:** 2
**Technical Novelty And Significance:** 2
**Empirical Novelty And Significance:** 2
**Recommendation:** 3

**Clarity, Quality, Novelty And Reproducibility:**

Some claims in the paper is not well explained. The method part could be further simplified to make it easier to follow. The method seems to be simple and reproducible.

**Strength And Weaknesses:**

Strength
1. The paper showed another way to achieve adaptive depth network, that is dropping layers in every stage rather than just the last stages/layers as in previous works.
2. The method achieves better performance than individually trained networks.

Weakness
1. I am not fully convinced by the claim that 'A block with a residual function F is supposed to perform two primary functions (1) learning new higher level features and (2) refining already learned features at the same semantic level'. The paper didn't provide many evidences. The only one maybe Fig.4. But to me, in Fig 4(b), I think the first three blocks didn't change much which the 3rd, 4th, 5th blocks changed a lot, which is against the author's claim.
2. In Table 1, the author only compare the proposed method with some very early works such as slimmable networks. I suggest the author to add some more recent works such as [1, 2]. It seems that the proposed method didn't outperform these works.
3. How did the author build the baseline models ResNet29 and ResNet35. These models are not used in the original paper.
4. The training cost should also be reported since it needs multiple forward pass.
5. I am not sure I understand how the proposed method works. The loss KL(h_{super}|h_{base}) will pull these two features close. The minimum loss will be reached when the skippable layers are optimized to be zero mapping. Thus I am not sure how the network learns.

Refereces

[1] Li, Changlin, et al. "Dynamic slimmable network." Proceedings of the IEEE/CVF Conference on Computer Vision and Pattern Recognition. 2021.
[2] Yang, Taojiannan, et al. "MutualNet: Adaptive ConvNet via Mutual Learning from Different Model Configurations." IEEE Transactions on Pattern Analysis and Machine Intelligence (2021).

**Summary Of The Paper:**

This paper proposed a method to train a network that can run at different depth at test time. The author claimed that a residual block could be considered to perform two functions (1) learning new features (2) refine features. The refinement stage/layers do not change feature semantics thus can be skipped. The proposed method is evaluated on multiple backbones and is demonstrated to achieve better performance than individually trained networks.

**Summary Of The Review:**

I am not fully convinced by the claims made in the paper. The experimental evaluation is also not complete and missing some references.

---

> ### Author Response · Authors · 2022-11-15
> **Response to Reviwer R1 (Part 2)**
>
>
> **R1-Q4**: How did the author build the baseline models ResNet29 and ResNet35. These models are not used in the original paper.
>
> **R1-A4**: ResNet29 and ResNet35, respectively, have the same network depths
> as ResNet50-ADN(baseline) and ResNet101-ADN(baseline).
> These models can be easily built by adjusting the number of block repetition
> of ResNet.
> For fair comparison, individual networks, including ResNet29 and ResNet35,
> are trained in the same training settings of their corresponding adaptive depth networks.
> We think that this model names can be confusing.
> Therefore, in the revision, we use ResNet50-Base instead of ResNet29.
> In Table 1, we explain that models with ‘-Base’ have the same depth as the base-net of corresponding adaptive depth network.
>
>
> **R1-Q5**: The training cost should also be reported since it needs multiple forward pass.
>
> **R1-A5**: Total training time is another important contribution of our work.
> Surely, training our adaptive depth networks take longer time than
> training a single network.
> Our work takes about 2 times longer training time than individual networks
> due to 2 forward passes.
> However, when we compare our work with other adaptive networks such as
> AlphaNet[1] and slimmable networks[3], our work has clear advantages.
> For instance, AlphaNet[1] and universal slimmable networks [3] both
> use 'sandwich rule' for training multiple sub-networks.
> With sandwich rule, at each training iteration,
> four sub-networks are sampled, or the largest, the smallest, and two random
> sub-networks.
> This means AlphaNet[1] uses all sub-networks during training.
> In contrast, our work uses only two sub-networks for training,
> or the super-net and the smallest base-net.
> In our work, other sub-networks are never explicitly used for training.
> However, as shown in Figure 3(a), other sub-networks can be used at test time
> and they provide inference accuracy proportional to their network depths.
> Therefore, in most adaptive networks, including AlphaNet[1], the total training time is linearly proportional to the number of supported sub-networks.
> In contrast, in our work, the total training time is no greater than training two sub-networks.
>
> In the revision, we emphasize this advantage in Introduction and Section 3.3.
>
> [1] Wang, Dilin, et al. "AlphaNet: improved training of supernets with alpha-divergence." International Conference on Machine Learning. PMLR, 2021\
> [2] Dosovitskiy, Alexey, et al. "An image is worth 16x16 words: Transformers for image recognition at scale." arXiv preprint arXiv:2010.11929 (2020).\
> [3] Yu et al. "Universally slimmable networks and improved training techniques." ICCV, 2019.
>
>
> **R1-Q6**:I am not sure I understand how the proposed method works. The loss $D_{KL}(\textbf{h}\_{super}|\textbf{h}\_{base})$ will pull these two features close. The minimum loss will be reached when the skippable layers are optimized to be zero mapping. Thus I am not sure how the network learns.
>
> **R1-A6**: Thanks for pointing this issue!
> In the revision, we update Section 3 extensively to discuss this more clearly.
> In summary, we show the loss function in Equation (8)
> to emphasize that the $D_{KL}(\textbf{h}\_{super}||\textbf{h}\_{base})$ terms are used as an regularization
> term in the loss function.
> Since $D_{KL}(\textbf{h}\_{super}||\textbf{h}\_{base})$ is not the only term in the loss function,
> $D_{KL}(\textbf{h}\_{super}||\textbf{h}\_{base})$ approaches to a small number, but it never becomes zero
> during training.
> The strength of $D_{KL}(\textbf{h}\_{super}||\textbf{h}\_{base})$ term is controlled by the hyperparameter $\alpha$.
> With this regularization term in the loss function, we can expect two effects:
> 1) Minimizing the KL terms make two features representations $\textbf{h}\_{base}$ and $\textbf{h}\_{super}$ have similar distributions over input $\textbf{X}$.
> This is important to skip a sub-path without causing sudden changes of input distribution to the next residual stage.
> 2)  Minimizing the KL terms also have
> the distillation effect of transferring the knowledge from $\textbf{h}\_{super}$
> to $\textbf{h}\_{base}$. Therefore, at each residual stage,
> $\textbf{h}\_{base}$ is expected to learn more compact
> representation from $\textbf{h}\_{super}$.

---

> ### Author Response · Authors · 2022-11-15
> **Response to Reviewer R1 (Part 1)**
>
> Thanks to reviewer R1 for helping improve our paper! We address the concerns as follows:
>
> **R1-Q1**: I am not fully convinced by the claim that 'A block with a residual function F is supposed to perform two primary functions (1) learning new higher level features and (2) refining already learned features at the same semantic level'.
>
> **R1-A1**:
> We acknowledge that our original sentences made some misunderstanding.
> In the revision, we rephrase as follows:
> *'While a block with a residual function F learns higher level features as
> traditional compositional networks, e.g., VGG networks, previous literature [1][2]
> demonstrates that a residual function also tend to learn a function
> that refines already learned features at the same feature level.'*
>
> [1] Jastrzebski et al. "Residual connections encourage iterative inference." ICLR, 2018\
> [2] Greff et al. "Highway and residual networks learn unrolled iterative estimation." ICLR, 2016
>
>
> **R1-Q2**: The paper didn't provide many evidences. The only one maybe Fig.4. But to me, in Fig 4(b), I think the first three blocks didn't change much which the 3rd, 4th, 5th blocks changed a lot, which is against the author's claim.
>
> **R1-A2**: We acknowledge that, in the original submission,
> we made unnecessary bold conjectures, such as *'suppressing one function in one part encourages the other part to focus on learning the suppressed function.'*
> We parallelized this to functional specialization in neuroscience.
> We acknowledge that such statements are not necessary in our work.
> In the revision, we extensively update Section 3.1 and 3.2 to remove
> such unnecessary claim.
>
> In Figure 4(b), we show that ResNet50-ADN (super-net)
> manifests very different progressions of activation regions in two sub-paths.
> With this we show that these two sub-paths are focusing on different
> tasks. The first 3 blocks have extensive hot activations in wide areas,
> and this implies active learning of features.
> In contrast, the skippable last 3 blocks have far less activation regions and they are gradually refined around the target.
> In the revision, we rephrase Section 4.4 to include above discussion.
>
> **R1-Q3**: In Table 1, the author only compare the proposed method with some very early works such as slimmable networks. I suggest the author to add some more recent works such as [1, 2]. It seems that the proposed method didn't outperform these works.
>
> **R1-A3**: We are very grateful for this suggestion!
> Unlike previous adaptive networks, our sub-paths specialization exploits the property of residual networks, and, hence, it is general to both CNNs and vision transformers (ViTs).
> As suggested, we apply our method to ViT to demonstrate its generality and novelty.
> We are extremely happy to report our evaluation results in Table 1. (Evaluation details on ViT can be found in the revised Section 4.)
>
> model | FLOPS |Acc@1 | Acc@5 | latency
> ---- | ----  | ----- | ------| ----
> **ViT-b/32-ADN (super-net)** |  2.95G | 76.4\%  |92.7\% | 39.5ms
> **ViT-b/32-ADN (base-net)** |  2.00G | 73.2\% | 91.4\% | 26.7ms
> ViT-b/32 (individual network) |2.95G | 75.9\% | 92.5\% | 39.5ms
> ViT-b/32-Base (individual network) | 2.00G | 73.1\% | 90.8\% | 26.7ms
>
> Even though we did minimal hyperparameter tuning due to limited computing resources,
> our results with ViT outperform individual networks' performance.
> (We only reduce the batch size and LR to half in PyTorch's standard training scripts.)
>
> **To the best of authors' knowledge, our work is the first general approach to
> adaptive networks demonstrating consistent performance improvements for both CNNs and vision transformers.**
>
> Further, in the revision, we compare our work with more recent state-of-the-art methods
> in Table 1 and Table 2. They include:
>
> [1] Wang, Dilin, et al. "AlphaNet: improved training of supernets with alpha-divergence." International Conference on Machine Learning. PMLR, 2021 \
> [2] Dosovitskiy, Alexey, et al. "An image is worth 16x16 words: Transformers for image recognition at scale." arXiv preprint arXiv:2010.11929 (2020). \
> [3] Yang et al. "Mutualnet: Adaptive convnet via mutual learning form network width and resolution." ECCV, 2020 \
> [4] Li et al. "Dynamic slimmable network." CVPR, 2021

---

### Author Response · Authors · 2022-11-15
**General Response**

We thank all the reviewers for their careful consideration of our paper and helpful suggestions.
We have updated our paper with a revised version that contains an improved exposition of our
work thanks to the reviewers.
In particular, we have done

**(1) Showing generality (and novelty) using new experiments with vision transformers**

Unlike previous adaptive networks, the proposed sub-paths specialization exploits the property of residual networks, and, hence, it is general to both CNNs and vision transformers (ViTs).
As suggested by several reviewers, we apply our method to ViT to demonstrate its generality.
We are extremely happy to report our evaluation results in Table 1.
(Evaluation details on ViT can be found in the revised Section 4.)

model | FLOPS |Acc@1 | Acc@5 | latency
---- | ----  | ----- | ------| ----
**ViT-b/32-ADN (super-net)** |  2.95G | 76.4\% |92.7\% | 39.5ms
**ViT-b/32-ADN (base-net)** |  2.00G | 73.2\% | 91.4\% | 26.7ms
ViT-b/32 (individual network) |2.95G | 75.9\% | 92.5\% | 39.5ms
ViT-b/32-Base (individual network) | 2.00G | 73.1\% | 90.8\% |26.7ms

Even though we did minimal hyperparameter tuning due to limited computing resources,
our results with ViT outperform individual networks' performance.
(We only reduce the batch size and LR  to half in PyTorch's standard training scripts.)

*To the best of authors' knowledge, our work is **the first general approach to
adaptive networks** demonstrating consistent performance improvements **for both CNNs and vision transformers.***

**(2) Clarifying theoretic parts (Section 3)**

Equations (2)-(5) in the original submission were
redundant and caused confusion since they did not match exactly
the implementation in Algorithm (1).
As suggested by reviewers, we have updated these equations.
The most important one is Equation (2) that is updated to:

$D_{KL} (\textbf{h}^s_{super} || \textbf{h}^s_{base}) \to 0$

This Equation matches our implementation and shows
more clearly why our approach improves the performance
while many sub-networks share parameters in a single model.
By having Equation (2) as an regularization term in the loss function,
as shown in Algorithm (1),
we can expect two effects:
1) Minimizing the KL terms make two features representations $\textbf{h}\_{base}$ and $\textbf{h}\_{super}$ have similar distributions over input $\textbf{X}$.
This is important to skip a sub-path without causing sudden changes of input distribution to the next residual stage.
2)  Minimizing the KL terms also have
the distillation effect of transferring the knowledge from $\textbf{h}\_{super}$
to $\textbf{h}\_{base}$. Therefore, at each residual stage,
$\textbf{h}\_{base}$ is expected to learn more compact
representation from $\textbf{h}\_{super}$.

Another important addition is Equation (7):
$F_{j+1}(\textbf{h}^{s}_{j}) \simeq  - \frac{\partial \mathcal{L}(  \textbf{h}^{s}_j ) }{\partial \textbf{h}^{s}_j}$.
Some reviewers questioned about the implication of Taylor expansion in the sub-paths specialization.
We believe that our original discussion was not clear enough.
We add Equation (7) to show more clearly that *residual functions in
the skippable sub-paths are optimized to learn a function
that has a similar effect to gradient descent.*
In other words, the residual functions in the skippable sub-paths
reduce the loss during inference.

**(3) Comparing our work with recent works (Experiments section)**

For fair comparison, our adaptive networks and
counterpart individual networks are trained in the same training settings.
Unlikely, several competing approaches to adaptive networks combine
some training techniques that are known to be very effective.
For instance, [1] exploits AutoAugment and [2] varies input resolutions during training.
*Our approach still achieves similar performance without such bells and whistles.*
In Table (1), we compare these recent competing approaches with ours for reference and mention above discussion.

Further, in Table (2), we compare our sub-networks with
state-of-the-art efficient inference methods.
They include (1) *static pruning methods*, (2) *new filter design*, and (3) *dynamic networks*.
These networks are either static networks or do not give users the freedom to control networks' resource consumption.
Unlikely, our sub-networks share parameters in a single model
and can switch to different sub-networks at runtime.
*Nevertheless, our sub-networks still achieve better or similar performance.*
To make this clear, we classify competing efficient inference methods
and annotate them in Table (2).


[1] Wang, Dilin, et al. "AlphaNet: improved training of supernets with alpha-divergence." International Conference on Machine Learning. PMLR, 2021 \
[2] Yang et al. "Mutualnet: Adaptive convnet via mutual learning form network width and resolution." ECCV, 2020

---

### Decision · Program_Chairs · 2023-01-20

**Decision:**

Reject

**Justification For Why Not Higher Score:**

All four expert reviewers consider the paper does not pass the acceptance bar of ICLR, unfortunately. The AC agrees with this decision. While the problem addressed by the authors is very important, the reviewers raised concerns about incremental novelty and outdated baselines. The AC agrees the method is similar to AlphaNet as pointed out by one of the reviewers. The author response provided new experiments, including more recent baselines, such as ViT. However, considering inference efficiency, as opposed to training, it is not clear that the proposed approach has significant advantages over AlphaNet and other approaches. It would also be desirable to see experiments with more modern ViT architectures (and shoe curves depicting latency vs accuracy at multiple operating points). The authors are encouraged to revise this work and submit the paper to another top conference.

**Justification For Why Not Lower Score:**

N/A

**Metareview: Summary, Strengths And Weaknesses:**

The paper introduces a method for anytime networks with adaptive depth without relying on intermediate gates/classifier decisions. The problem addressed by the authors is very important and timely. However, all reviewers raised major concerns about the paper, including incremental novelty over prior methods (e.g., AlphaNet) and outdated baselines.

**Summary Of Ac-Reviewer Meeting:**

N/A